# ATTENTION INTERPRETABILITY ACROSS NLP TASKS

## ABSTRACT

The attention layer in a neural network model provides insights into the model's reasoning behind its prediction, which are usually criticized for being opaque. Recently, seemingly contradictory viewpoints have emerged about the interpretability of attention weights (Jain & Wallace, 2019; Vig & Belinkov, 2019). Amid such confusion arises the need to understand attention mechanism more systematically. In this work, we attempt to fill this gap by giving a comprehensive explanation which justifies both kinds of observations (i.e., when is attention interpretable and when it is not). Through a series of experiments on diverse NLP tasks, we validate our observations and reinforce our claim of interpretability of attention through manual evaluation.

## 1 INTRODUCTION

Attention is a way of obtaining a weighted sum of the vector representations of a layer in a neural network model (Bahdanau et al., 2015). It is used in diverse tasks ranging from machine translation (Luong et al., 2015), language modeling (Liu & Lapata, 2018) to image captioning (Xu et al., 2015), and object recognition (Ba et al., 2014). Apart from substantial performance benefit (Vaswani et al., 2017), attention also provides interpretability to neural models (Wang et al., 2016; Lin et al., 2017; Ghaeini et al., 2018) which are usually criticized for being black-box function approximators (Chakraborty et al., 2017).

There has been substantial work on understanding attention in neural network models. On the one hand, there is work on showing that attention weights are not interpretable, and altering them does not significantly affect the prediction (Jain & Wallace, 2019; Serrano & Smith, 2019). While on the other hand, some studies have discovered how attention in neural models captures several linguistic notions of syntax and coreference (Vig & Belinkov, 2019; Clark et al., 2019; Tenney et al., 2019). Amid such contrasting views arises a need to understand the attention mechanism more systematically. In this paper, we attempt to fill this gap by giving a comprehensive explanation which justifies both kinds of observations.

The conclusions of Jain & Wallace (2019); Serrano & Smith (2019) have been mostly based on text classification experiments which might not generalize to several other NLP tasks. In Figure 1, we report the performance on text classification, Natural Language Inference (NLI) and Neural Machine Translation (NMT) of two models: one trained with neural attention and the other trained with attention weights fixed to a uniform distribution. The results show that the attention mechanism in text classification does not have an impact on the performance, thus, making inferences about interpretability of attention in these models might not be accurate. However, on tasks such as NLI and NMT uniform attention weights degrades the performance substantially, indicating that attention is a crucial component of the model for these tasks and hence the analysis of attention's interpretability here is more reasonable.

In comparison to the existing work on interpretability, we analyze attention mechanism on a more diverse set of NLP tasks that include text classification, pairwise text classification (such as NLI), and text generation tasks like neural machine translation (NMT). Moreover, we do not restrict ourselves to a single attention mechanism and also explore models with self-attention. For examining the interpretability of attention weights, we perform manual evaluation. Our key contributions are:

1. We extend the analysis of attention mechanism in prior work to diverse NLP tasks and provide a comprehensive picture which alleviates seemingly contradicting observations.

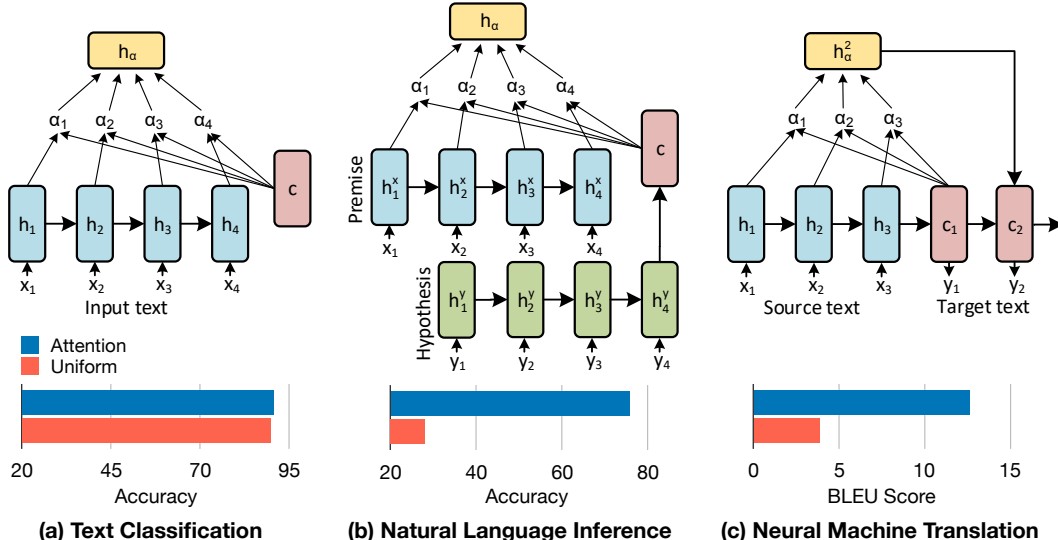

Figure 1: Comparison of performance with and without neural attention on text classification (IMDB), Natural Language Inference tasks (SNLI) and Neural Machine Translation (News Commentary). Here, $\alpha$ and $c$ denote attention weights and context vector respectively. The results show that attention does not substantially effect performance on text classification. However, the same does not hold for other tasks.

2. We identify the conditions when attention weights are interpretable and correlate with feature importance measures – when they are computed using two vectors which are both functions of the input (Figure 1b, c). We also explain why attention weights are not interpretable when the input has only single sequence (Figure 1a), an observation made by Jain & Wallace (2019), by showing that they can be viewed as a gating unit.

3. We validate our hypothesis of interpretability of attention through manual evaluation.

## 2 TASKS AND DATASETS

We investigate the attention mechanism on the following three task categories.

1. **Single Sequence tasks** are those where the input consists of a single text sequence. For instance, in sentiment analysis, the task is to classify a review as positive or negative. This also includes other text classification tasks such as topic categorization. For the experiments, in this paper, we use three review rating datasets: (1) Stanford Sentiment Treebank (Socher et al., 2013), (2) IMDB (Maas et al., 2011) and (3) Yelp 2017[1] and one topic categorization dataset AG News Corpus *(business vs world)*.[2]

2. **Pair Sequence tasks** comprise of a pair of text sequences as input. The tasks like NLI and question answering come under this category. NLI involves determining whether a *hypothesis* entails, contradicts, or is undetermined given a *premise*. We use Stanford Natural Language Inference (SNLI) (Bowman et al., 2015) and Multi-Genre Natural Language Inference (MultiNLI) (Williams et al., 2018) datasets for our analysis. For question answering, similar to Jain & Wallace (2019), we use CNN News Articles (Hermann et al., 2015) and three tasks of the original babI dataset (Weston et al., 2015) in our experiments, i.e., using one, two and three supporting statements as the context for answering the questions.

3. **Generation tasks** involve generating a sequence based on the input sequence. Neural Machine translation is an instance of generation task which comprises of translating a source text to a target language given translation pairs from a parallel corpus. For our experiments, we use

---

[1]from www.yelp.com/dataset_challenge
[2]www.di.unipi.it/ gulli/AG_corpus_of_news_articles.html

three English-German datasets: Multi30k (Elliott et al., 2016), En-De News Commentary v11 from WMT16 translation task[3] and full En-De WMT13 dataset.

## 3 NEURAL ATTENTION MODELS

In this section, we give a brief overview of the neural attention-based models we analyze for different categories of tasks listed in Section 2. The overall architecture for each category is shown in Fig 1.

### 3.1 SINGLE SEQUENCE MODELS:

For single sequence tasks, we adopt the model architecture from Jain & Wallace (2019); Wiegreffe & Pinter (2019). For a given input sequence $\boldsymbol{x} \in \mathbb{R}^{T \times |V|}$, where $T$ and $|V|$ are the number of tokens and vocabulary size, we first represent each token with its $d$-dimensional GloVe embedding Pennington et al. (2014) to obtain $\boldsymbol{x}_e \in \mathbb{R}^{T \times d}$. Next, we use a Bi-RNN encoder (**Enc**) to obtain an $m$-dimensional contextualized representation of tokens: $\boldsymbol{h} = \mathbf{Enc}(\boldsymbol{x}_e) \in \mathbb{R}^{T \times m}$. Then, we use the additive formulation of attention proposed by Bahdanau et al. (2015) for computing attention weights $\alpha_i$ for all tokens defined as:

$$\boldsymbol{u}_i = \tanh(\boldsymbol{W}\boldsymbol{h}_i + \boldsymbol{b}); \quad \alpha_i = \frac{\exp(\boldsymbol{u}_i^T \boldsymbol{c})}{\sum_j \exp(\boldsymbol{u}_j^T \boldsymbol{c})}, \tag{1}$$

where $\boldsymbol{W} \in \mathbb{R}^{d' \times m}, \boldsymbol{b}, \boldsymbol{c} \in \mathbb{R}^{d'}$ are the parameters of the model. Finally, the weighted instance representation: $\boldsymbol{h}_\alpha = \sum_{i=1}^{T} \alpha_i \boldsymbol{h}_i \in \mathbb{R}^m$ is fed to a dense layer (**Dec**) followed by softmax to obtain prediction $\hat{y} = \sigma(\mathbf{Dec}(\boldsymbol{h}_\alpha)) \in \mathbb{R}^{|\mathcal{Y}|}$, where $|\mathcal{Y}|$ denotes the label set size.

We also analyze the hierarchical attention model (Yang et al., 2016), which involves first computing attention over the tokens to obtain a sentence representation. This is followed by attention over sentences to obtain an instance representation $\boldsymbol{h}_\alpha$, which is fed to a dense layer for obtaining prediction $(\hat{y})$. At both word and sentence level the attention is computed similar to as defined in Equation 1.

### 3.2 PAIR SEQUENCE MODELS:

For pair sequence, the input consists of two text sequences: $\boldsymbol{x} \in \mathbb{R}^{T_1 \times |V|}, \boldsymbol{y} \in \mathbb{R}^{T_2 \times |V|}$ of length $T_1$ and $T_2$. In NLI, $\boldsymbol{x}$ indicates premise and $\boldsymbol{y}$ is hypothesis while in question answering, it is the question and paragraph respectively. Following Bowman et al. (2015), we use two separate RNNs for encoding both the sequences to obtain $\{\boldsymbol{h}_1^x, ..., \boldsymbol{h}_{T_1}^x\}$ and $\{\boldsymbol{h}_1^y, ..., \boldsymbol{h}_{T_2}^y\}$. Now, similar to Jain & Wallace (2019), attention weight $\alpha_i$ over each token of $\boldsymbol{x}$ is computed as:

$$u_i = \tanh(\boldsymbol{W}_1 \boldsymbol{h}_i^x + \boldsymbol{W}_2 \boldsymbol{h}_{T_2}^y); \quad \alpha_i = \frac{\exp(\boldsymbol{u}_i^T \boldsymbol{c})}{\sum_j \exp(\boldsymbol{u}_j^T \boldsymbol{c})}, \tag{2}$$

where similar to Equation 1, $\boldsymbol{W}_1, \boldsymbol{W}_2 \in \mathbb{R}^{d \times d'}$ denotes the projection matrices and $\boldsymbol{c} \in \mathbb{R}^{d'}$ is a parameter vector. Finally, the representation obtained from a weighted sum of tokens in $\boldsymbol{x}$: $\boldsymbol{h}_\alpha = \sum_{i=1}^{T} \alpha_i \boldsymbol{h}_i^x$ is fed to a classifier for prediction.

We also explore a variant of the above attention proposed by Rocktäschel et al. (2016). Instead of keeping the RNN encoders of both the sequences independent, Rocktäschel et al. (2016) use conditional encoding where the encoder of $\boldsymbol{y}$ is initialized with the final state of $\boldsymbol{x}$'s encoder. This allows the model to obtain a conditional encoding $\{\boldsymbol{h}_1'^y, ..., \boldsymbol{h}_{T_2}'^y\}$ of $\boldsymbol{y}$ given the sequence $\boldsymbol{x}$. Moreover, unlike the previous model, attention over the tokens of $\boldsymbol{x}$ is defined as follows:

$$\boldsymbol{M} = \tanh(\boldsymbol{W}_1 \boldsymbol{X} + \boldsymbol{W}_2 \boldsymbol{h}_{T_2}'^y \otimes \boldsymbol{e}_{T_1}); \quad \boldsymbol{\alpha} = \mathrm{softmax}(\boldsymbol{w}^T \boldsymbol{M}), \tag{3}$$

where $\boldsymbol{X} = [\boldsymbol{h}_1^x, ..., \boldsymbol{h}_{T_1}^x]$, $\boldsymbol{e}_{T_1} \in \mathbb{R}^{T_1}$ is a vector of ones and outer product $\boldsymbol{W}_2 \boldsymbol{h}_{T_2}'^y \otimes \boldsymbol{e}_{T_1}$ denotes repeating linearly transformed $\boldsymbol{h}_{T_2}'^y$ as many times as words in the sequence $\boldsymbol{x}$ (i.e. $T_1$ times).

---

[3]http://www.statmt.org/wmt16/translation-task.html

### 3.3 GENERATION TASK MODELS:

In this paper, for generation tasks, we focus on Neural Machine Translation (NMT) problem which involves translating a given source text sentence $\boldsymbol{x} \in \mathbb{R}^{T_1 \times |V_1|}$ to a sequence $\boldsymbol{y} \in \mathbb{R}^{T_2 \times |V_2|}$ in the target language. The model comprises of two components: (a) an encoder which computes a representation for each source sentence and (b) a decoder which generates a target word at each time step. In this work, we utilize RNN based encoder and decoder models. For each input sentence $\boldsymbol{x}$, we first obtain a contextualized representation $\{\boldsymbol{h}_1, ..., \boldsymbol{h}_{T_1}\}$ of its tokens using a multi-layer Bi-RNN. Then, at each time step $t$, the decoder has a hidden state defined as

$$\boldsymbol{c}_t = f(\boldsymbol{c}_{t-1}, y_{t-1}, \boldsymbol{h}_\alpha^t), \text{where } \boldsymbol{h}_\alpha^t = \sum_{i=1}^{T_1} \alpha_{t,i} \boldsymbol{h}_i.$$

In our work, we compute $\alpha_{t,i}$ as proposed by Bahdanau et al. (2015) and Luong et al. (2015). The former computes attention weights using a feed-forward network, i.e., $\alpha_{t,i} = \boldsymbol{w}^T \tanh(\boldsymbol{W}[\boldsymbol{c}_t; h_i])$ while the latter define it simply as $\alpha_{t,i} = \boldsymbol{c}_t^T h_i$.

### 3.4 SELF-ATTENTION BASED MODELS:

We also examine *self-attention* based models on all three categories of tasks. For single and pair sequence tasks, we fine-tune pre-trained BERT (Devlin et al., 2019) model on the downstream task. In pair sequence tasks, instead of independently encoding each text, we concatenate both separated by a delimiter and pass it to BERT model. Finally, the embedding corresponding to `[CLS]` token is fed to a feed-forward network for prediction. For neural machine translation, we use Transformer model proposed by Vaswani et al. (2017) with *base* configuration.

## 4 IS ATTENTION AN EXPLANATION?

In this section, we attempt to address the question: *Is attention an explanation?* through a series of experiments which involve analyzing attention weights in a variety of models (§3) on multiple tasks (§2). Following Jain & Wallace (2019), we take the definition of *explainability* of attention as: inputs with high attention weights are responsible for model output. Jain & Wallace (2019); Serrano & Smith (2019) have extensively investigated this aspect for certain class of problems and have shown that attention does not provide an explanation. However, another series of work (Vig & Belinkov, 2019; Clark et al., 2019; Tenney et al., 2019) has shown that attention does encode several linguistic notions. In our work, we claim that the findings of both the line of work are consistent. We note that the observations of the former works can be explained based on the following proposition.

**Proposition 4.1.** *Attention mechanism as defined in Equation 1 as*

$$\boldsymbol{u}_i = \tanh(\boldsymbol{W}\boldsymbol{h}_i + \boldsymbol{b}); \quad \alpha_i = \frac{\exp(\boldsymbol{u}_i^T \boldsymbol{c})}{\sum_j \exp(\boldsymbol{u}_j^T \boldsymbol{c})}$$

*for single sequence tasks can be interpreted as a gating unit in the network.*

**Proof:** The attention weighted averaging computed in Equation 1 for single sequence tasks can be interpreted as gating proposed by Dauphin et al. (2017) which is defined as

$$\boldsymbol{h}(\boldsymbol{x}) = f(\boldsymbol{x}) \times \sigma(g(\boldsymbol{x})),$$

where $\boldsymbol{x} \in \mathbb{R}^m$ is the input and $\times$ denotes product between transformed input $f(\boldsymbol{x}) \in \mathbb{R}^m$ and its computed gating scores $\sigma(g(\boldsymbol{x})) \in \mathbb{R}$. Equation 1 can be reduced to the above form by taking $f$ as an identity function and defining $g(\boldsymbol{x}) = \boldsymbol{c}^T \tanh(\boldsymbol{W}\boldsymbol{x} + \boldsymbol{b}) \in \mathbb{R}$ and replacing $\sigma$ with softmax. We note that the same reduction does not hold in the case of pair sequence and generation tasks where attention along with input also depends on another text sequence $\boldsymbol{Y}$ and current hidden state $\boldsymbol{c}_t$, respectively. Thus, attention mechanism for these tasks take the form

$$\boldsymbol{h}(\boldsymbol{x}, \boldsymbol{y}) = f(\boldsymbol{x}) \times \sigma(g(\boldsymbol{x}, \boldsymbol{y})),$$

which does not reduce to the above equation for gating unit. □

|  | SST | IMDB | AG News | YELP |
|---|---|---|---|---|
| Bahdanau et al. (2015) | $83.4 \pm 0.5$ | $90.7 \pm 0.7$ | $96.4 \pm 0.1$ | $66.7 \pm 0.1$ |
| Uniform (Train+Infer / Infer) | $-1.0\,/-0.8$ | $-0.8\,/-6.3$ | $-0.1\,/-0.7$ | $-0.5\,/-6.3$ |
| Random (Train+Infer / Infer) | $-1.1\,/-0.9$ | $-0.6\,/-6.4$ | $-0.0\,/-0.7$ | $-0.4\,/-6.4$ |
| Permute (Infer) | $-1.7$ | $-5.1$ | $-0.9$ | $-7.8$ |
| Yang et al. (2016) | $83.2 \pm 0.5$ | $89.7 \pm 0.6$ | $96.1 \pm 0.2$ | $65.8 \pm 0.1$ |
| Uniform (Train+Infer / Infer) | $-1.0\,/-0.8$ | $+0.2\,/-6.5$ | $+0.1\,/-1.5$ | $-0.7\,/-8.0$ |
| Random (Train+Infer / Infer) | $-0.9\,/-1.0$ | $-1.2\,/-8.2$ | $-0.1\,/-1.8$ | $-3.0\,/-10.2$ |
| Permute (Infer) | $-1.8$ | $-5.1$ | $-0.7$ | $-10.7$ |

Table 1: Evaluation results on single sequence tasks. We report the base performance of attention models and absolute change in accuracy for its variant. We note that across all datasets, degradation in performance on altering attention weights during inference is more compared to varying them during both training and inference. Overall, the change in performance is less compared to other tasks. Please refer to §4.1 for more details.

Based on the above proposition, we argue that weights learned in single sequence tasks cannot be interpreted as attention, and therefore, they do not reflect the reasoning behind the model's prediction. This justifies the observation that for the single sequence tasks examined in Jain & Wallace (2019); Serrano & Smith (2019), attention weights do not correlate with feature importance measures and permuting them does not change the prediction of the model. In light of this observation, we revisit the explainability of attention weights by asking the following questions.

## 4.1 How does altering attention weights affect model output on tasks?

In this section, we compare the performance of various attention mechanism described in §3 for different categories of tasks listed in §2. For each model, we analyze its three variants defined as:

- **Uniform** denotes the case when all the inputs are given equal weights, i.e., $\alpha_i = 1/T$, $\forall i \in \{1, ..., T\}$. This is similar to the analysis performed by Wiegreffe & Pinter (2019). However, we consider two scenarios when the weights are kept fixed both during training and inference (Train+Infer) and only during inference (Infer).

- **Random** refers to the variant where all the weights are randomly sampled from a uniform distribution: $\alpha_i \sim U(0, 1)$, $\forall i \in \{1, ..., T\}$, this is followed by normalization. Similar to Uniform, we analyze both Train+Infer and Infer.

- **Permute** refers to the case when the learned attention weights are randomly permuted during inference, i.e., $\boldsymbol{\alpha} = \text{shuffle}(\boldsymbol{\alpha})$. Unlike the previous two, here we restrict our analysis to only permuting during inference as Tensorflow currently does not support backpropagation with shuffle operation.[4]

**Effect on single sequence tasks:** The evaluation results on single sequence datasets: SST, IMDB, AG News, and YELP are presented in Table 1. We observe that Train+Infer case of Uniform and Random attentions gives around $0.5$ and $0.9$ average decrease in accuracy compared to the base model. However, in Infer scenario the degradation on average increases to $3.9$ and $4.5$ absolute points respectively. This is so because the model becomes more robust to handle altered weights in the former case. The reduction in performance from Permute comes around to $4.2$ across all datasets and models. The results support the observation of Jain & Wallace (2019); Serrano & Smith (2019) that alternating attention in text classification task does not have much effect on the model output. The slight decrease in performance can be attributed to corrupting the existing gating mechanism which has been shown to give some improvement (Oord et al., 2016; Dauphin et al., 2017; Marcheggiani & Titov, 2017).

**Effect on pair sequence and generation tasks:** The results on pair sequence and generation tasks are summarized in Table 2 and 3, respectively. Overall, we find that the degradation in performance from altering attention weights in case of pair sequence and generation tasks is much more substantial than single sequence tasks. For instance, in Uniform (Train+Infer), the average relative decrease

---

[4]https://github.com/tensorflow/tensorflow/issues/6269

|  | **SNLI** | **MultiNLI** | **CNN** | **babI 0** | **babI 1** | **babI 2** |
|---|---|---|---|---|---|---|
| Bahdanau et al. (2015) | $75.7 \pm 0.3$ | $61.1 \pm 0.1$ | $63.4 \pm 0.8$ | $96.1 \pm 4.3$ | $95.8 \pm 0.3$ | $92.8 \pm 0.1$ |
| Uniform (Train+Infer / Infer) | $-41.8\,/\,-42.9$ | $-26.6\,/\,-28.7$ | $-30.8\,/\,-55.9$ | $-44.4\,/\,-63.4$ | $-47.4\,/\,-60.4$ | $-48.4\,/\,-62.1$ |
| Random (Train+Infer / Infer) | $-41.6\,/\,-43.1$ | $-26.7\,/\,-28.6$ | $-30.9\,/\,-55.9$ | $-45.0\,/\,-62.0$ | $-47.3\,/\,-60.4$ | $-49.9\,/\,-62.2$ |
| Permute (Infer) | $-41.0$ | $-27.6$ | $-54.8$ | $-67.8$ | $-68.3$ | $-66.7$ |
| Rocktäschel et al. (2016) | $78.1 \pm 0.2$ | $62.4 \pm 0.6$ | $63.6 \pm 0.6$ | $98.6 \pm 1.6$ | $96.2 \pm 0.9$ | $93.2 \pm 0.1$ |
| Uniform (Train+Infer / Infer) | $-44.2\,/\,-45.4$ | $-27.5\,/\,-30.3$ | $-30.8\,/\,-43.1$ | $-47.7\,/\,-67.8$ | $-47.9\,/\,-62.8$ | $-49.8\,/\,-60.9$ |
| Random (Train+Infer / Infer) | $-44.3\,/\,-44.9$ | $-27.9\,/\,-28.3$ | $-30.6\,/\,-43.3$ | $-47.5\,/\,-64.9$ | $-48.4\,/\,-63.3$ | $-49.8\,/\,-60.9$ |
| Permute (Infer) | $-41.7$ | $-29.2$ | $-44.9$ | $-68.8$ | $-68.3$ | $-65.2$ |

Table 2: The performance comparison of attention based models and their variants on pair sequence tasks. We find that the degradation in performance is much more than single sequence tasks.

| Dataset | **Multi30k** | **News Commentary** |
|---|---|---|
| Bahdanau et al. (2015) | $31.3 \pm 0.1$ | $12.6 \pm 0.1$ |
| Uniform (Train+Infer / Infer) | $-10.4\,/\,-29.4$ | $-8.7\,/\,-11.8$ |
| Random (Train+Infer / Infer) | $-10.1\,/\,-29.4$ | $-8.8\,/\,-11.9$ |
| Permute (Infer) | $-29.7$ | $-12.1$ |
| Luong et al. (2015) | $31.5 \pm 0.2$ | $12.7 \pm 0.2$ |
| Uniform (Train+Infer / Infer) | $-10.6\,/\,-29.7$ | $-8.8\ /\,-12.0$ |
| Random (Train+Infer / Infer) | $-10.3\,/\,-29.8$ | $-8.9\ /\,-12.0$ |
| Permute (Infer) | $-30.1$ | $-12.2$ |

Table 3: Evaluation results on neural machine translation. Similar to pair sequence tasks, we find that the deterioration in performance is much more substantial than single sequence tasks. Please refer to §4.1 for more details.

in performance of single sequence tasks is $0.1\%$ whereas in case of pair sequence and generation tasks it is $49.5\%$ and $51.2\%$ respectively. The results thereby validate our Proposition 4.1 and show that altering attention does affect model output for a task where the attention layer cannot be modeled as a gating unit in the network.

**Visualizing the effect of permuting attention weights:** To further reinforce our claim, similar to Jain & Wallace (2019), we report the median of Total Variation Distance (TVD) between new and original prediction on permuting attention weights for each task. The TVD between two predictions $\hat{y}_1$ and $\hat{y}_2$ is defined as: $\mathrm{TVD}(\hat{y}_1, \hat{y}_2) = \frac{1}{2} \sum_{i=1}^{|\mathcal{Y}|} |\hat{y}_{1i} - \hat{y}_{2i}|$, where $|\mathcal{Y}|$ denotes the total number of classes in the problem. We use TVD for measuring the change in output distribution on permuting the attention weights. In Figure 2, we report the relationship between the maximum attention value and the median induced change in model output over 100 permutations on all categories of tasks. For NMT task, we present change in output at the 25th-percentile length of sentences for both datasets. Overall, we find that for single sequence tasks even with the maximum attention weight in range $[0.75, 1.0]$, the change in prediction is considerably small (the violin plots are to the left of the figure) compared to the pair sequence and generation tasks (the violin plots are to the right of the figure).

## 4.2 DO ATTENTION WEIGHTS CORRELATE WITH FEATURE IMPORTANCE MEASURES?

In this section, similar to the analysis of Serrano & Smith (2019), we investigate the importance of attention weights only when one weight is removed. Let $i^*$ be the input corresponding to the highest attention weights and let $r$ be any randomly selected input. We denote the original model's prediction as $p$ and output after removing $i^*$ and $r$ input as $q_{\{i^*\}}$ and $q_{\{r\}}$ respectively. Now, to measure the impact of removing $i^*$ relative to any randomly chosen input $r$ on the model output, we compute the difference of Jensen-Shannon (JS) divergence between $\mathrm{JS}(p, q_{\{i^*\}})$ and $\mathrm{JS}(p, q_{\{r\}})$ given as: $\Delta \mathrm{JS} = \mathrm{JS}(p, q_{\{i^*\}}) - \mathrm{JS}(p, q_{\{r\}})$. The relationship between the difference of attention weights corresponding to $i^*$ and $r$, i.e., $\alpha_{i^*} - \alpha_r$ and $\Delta \mathrm{JS}$ for different tasks is presented in Figure 3. In general, we found that for single sequence tasks, the change JS divergence is small even for cases

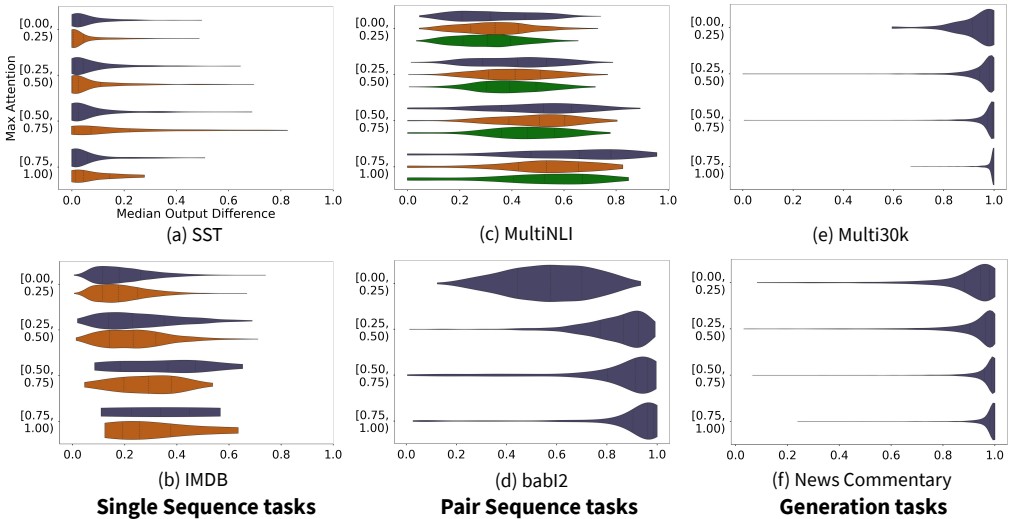

Figure 2: Relationship between maximum attention weight and median change in output on permuting attention weights. For single sequence tasks, ■, ■ indicate negative and positive class. For MultiNLI, ■, ■, ■ denotes contradiction, entailment and neutral respectively. The results reinforce the claim that altering attention weights in single sequence tasks does not have much effect on performance while the same does not hold with other tasks. Refer to §4.1 for details.

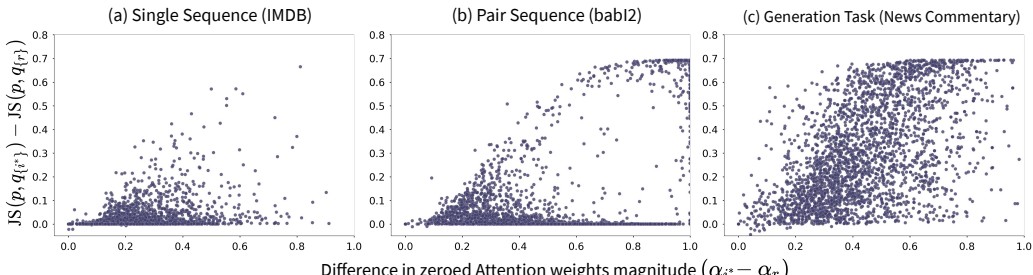

Figure 3: Analysis of correlation between attention weights and feature importance measure. We report relationship between difference in zeroed attention weights and corresponding change in $JS$ divergence for different tasks. Please refer to §4.2 for more details.

when the difference in attention weight is considerable. However, for pair sequence and generation tasks, there is a substantial change in the model output.

### 4.3 HOW PERMUTING DIFFERENT LAYERS OF SELF-ATTENTION BASED MODELS AFFECT PERFORMANCE?

In this section, we analyze the importance of attention weights on the performance of self-attention based models as described in §3.4. We report the accuracy on single, and pair sequence tasks and BLEU score for NMT on WMT13 dataset on permuting the attention weights of layers cumulatively. For Transformer model, we analyze the effect of altering attention weights in encoder, decoder, and across encoder-decoder (denoted by Across). The results are presented in Figure 4. Overall, we find that unlike the pattern observed in §4.1 and §4.2 for single sequence tasks, altering weights in self-attention based models does have a substantial effect on the performance. We note that this is because while computing attention weights over all tokens with respect to a given token, Proposition 4.1 does not hold. Thus, altering them does have an impact across all three tasks. We note that in the case of transformer model, altering the weights in the first step of Decoder and in Across has maximum effect as it almost stops the flow of information from encoder to decoder.

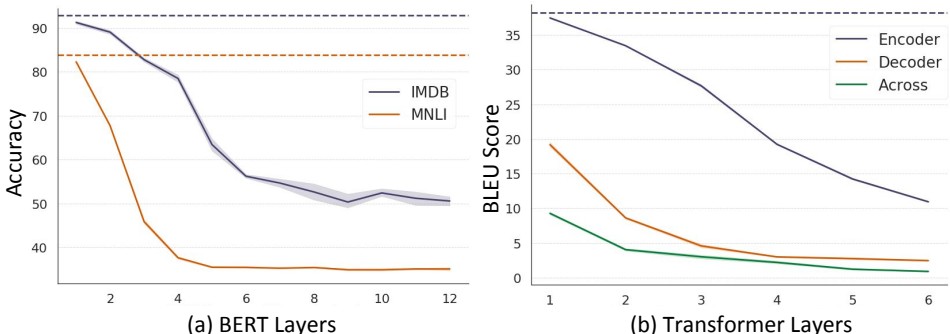

Figure 4: Performance comparison with permuting attention weights for different layers in self-attention based models. The results are reported on a representative instance of single sequence, pair sequence and generation tasks. The dotted lines denote the base performance on the task. Refer to §4.3 for details.

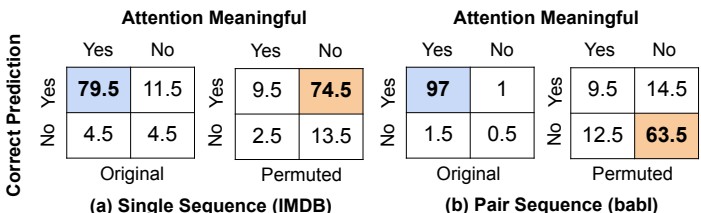

Figure 5: Manual evaluation of interpretability of attention weights on single and pair sequence tasks. Although with original weights the attention does remain interpretable on both tasks but in the case of single sequence tasks making it meaningless does not change the prediction substantially. However, the same does **not** hold with pair sequence tasks.

## 4.4 ARE ATTENTION WEIGHTS HUMAN INTERPRETABLE?

To determine if attention weights are human interpretable, here, we address the question of interpretability of attention weights by manually analyzing them on a representative dataset of single and pair sequence task. For each task, we randomly sample 100 samples with original attention weights and 100 with randomly permuted weights. Then, we shuffle all 200 samples together and present them to annotators for deciding whether the top three highest weighted words are relevant for the model's prediction.

The overall results are reported in Figure 5. Cohen's kappa score of inter-annotator agreement (Cohen, 1960) on IMDB and babI is $0.84$ and $0.82$, respectively, which shows near-perfect agreement (Landis & Koch, 1977). We find that in both single and pair sequence tasks, the attention weights in samples with original weights do make sense in general (highlighted with blue color). However, in the former case, the attention mechanism learns to give higher weights to tokens relevant to both kinds of sentiment. For instance, in "*This is a great movie. Too bad it is not available on home video.*", tokens *great*, *too*, and *bad* get the highest weight. Such examples demonstrate that the attention mechanism in single sequence tasks works like a gating unit, as shown in §4.1.

For permuted samples, in the case of single sequence, the prediction remains correct in majority although the attention weights were meaningless. For example, in "*This movie was terrible . the acting was lame , but it's hard to tell since the writing was so bad .*", the prediction remains the same on changing attention weights from underlined to **bold** tokens. However, this does not hold with the pair sequence task. This shows that attention weights in single sequence tasks do not provide a reason for the prediction, which in the case of pairwise tasks, attention do reflect the reasoning behind model output.

## 5  CONCLUSION

In this paper, we addressed the seemingly contradictory viewpoint over explainability of attention weights in NLP. On the one hand, some works have demonstrated that attention weights are not interpretable, and altering them does not affect the model output while several others have shown that attention captures several linguistic notions in the model. We extend the analysis of prior works to diverse NLP tasks and demonstrate that attention weights are interpretable and are correlated with feature importance measures. However, this holds only for cases when attention weights are essential for model's prediction and cannot simply be reduced to a gating unit. Through a battery of experiments, we validate our claims and reinforce them through manual evaluation.

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
