# OpenReview forum: "Attention Interpretability Across NLP Tasks"
_ICLR.cc/2020/Conference — Reject_

### Official Review · AnonReviewer1 · 2019-10-17
**Official Blind Review #1**

**Rating:** 6

**Review:**

Motivated by an existing paper, the paper analyzes the interpretability of attention mechanism over three NLP tasks. The previous paper claims that attention mechanism is not interpratable. The paper makes incremental contribution by showing that attentions are not interpratable when they mimic gating units (single sequence task) but are interpretable for two sequence and generation tasks. Experimental results are given to support the claims, which can also help readers to gain insights into the attention mechanism. The paper is well written and all claims are supported. I also have some questions below for clarification. If I get reasonable answers to these questions, I tend to accept the paper.

1. Jain & Wallace (2019) show that for the SNLI problem, attentions weakly correlate with the output based on Kendall's correlation and JSD which contradicts to your observation. Could you explain how this happens? are there any model settings different?
2. in figure 5, for the single sequence (original), most of attentions leading to correct predictions are labeled meaningful. Does this mean that even the attention doesn't necessarily contribute too much to the prediction correctness but they are also interpretable if they are allowed to be trained. Also, in Table 1, with modified attentions, all scores go down a little. This means that attention still can contribute to the final prediction but not significant enough (some like yelp are significant). I am gussing that the sequence encoding already present useful features for the final prediction. Did you check the distance of different word encodings? Are these encodings all very similar?

**Experience Assessment:**

I have read many papers in this area.

**Review Assessment: Checking Correctness Of Derivations And Theory:**

I carefully checked the derivations and theory.

**Review Assessment: Checking Correctness Of Experiments:**

I carefully checked the experiments.

**Review Assessment: Thoroughness In Paper Reading:**

I read the paper at least twice and used my best judgement in assessing the paper.

---

> ### Author Response · Authors · 2019-11-07
> **Thank you for the feedback!**
>
> Thank you for the constructive comments. We are glad that the reviewer liked our work. Below we provide clarification for the reviewer’s queries.
>
> 1. In comparison to Jain & Wallace (2019), we use the exact same model architecture, i.e., Bi-LSTM over word embedding of tokens followed by attention. Although for deciding hyperparameters we use validation data, unlike Jain & Wallace (2019) who use test data for that. In our experiments, we did not analyze how attention correlates with output based on Kendall and JSD but permutation scatterplots presented in Figure 2 are very similar to what has been reported by Jain & Wallace (2019) although they do not highlight them much in their paper. The complete set of results of Jain & Wallace (2019) are provided at this link: https://successar.github.io/AttentionExplanation/docs/. Here, we find that Permutation Scatterplots on Babi and SNLI datasets are very similar to what we have reported.
>
> 2. For manual evaluation, an annotator was asked to give a binary yes/no based on whether attention weights are meaningful or not. For that, an annotator is shown top three tokens with highest attention weights and if at least of them supports the prediction then “yes” was assigned otherwise “no”. For the case of single sequence tasks with original attention weights, we find that the gating mechanism during training learns to give high scores to tokens which refect some sentiment but does not reflect the reasoning behind the model’s prediction. An example of the same is also presented in Section 4.4. Since a given review has few tokens which directly highlight the sentiment so, with our evaluation setting at least one such token is highly likely to appear in the top three. But we can see that the weights do not influence the model output as changing them does not affect the prediction considerably as shown in Figure 5 (a) Permuted and by the series of papers [1, 2] prior to our work.
>
> Yes, attention even in the case of single sequence tasks is helpful as it is adding more parameters to the model and thus allowing it to learn more complex relationships between the input and the output. The size of datasets like YELP and IMDB is considerably large compared to other datasets, therefore providing more parameters allows the model to learn more from training data and perform better. This is reflected through significant changes in the performance on them.
>
> Although we have not explicitly performed such analysis but based on the variation in attention weights over tokens (see x-axis Figure 3(a)) we can say that the words encoding are not very similar. The attention weights in single sequence tasks for each token are computed using the same W and c parameters. Now, since the difference between a randomly selected attention weight and the highest attention weight is going up to 0.6 on IMDB, where average text length is around 282 words, based on this we believe that the word encoding should not be very similar.
>
> [1] Jain, Sarthak and Byron C. Wallace. “Attention is not Explanation.” NAACL-HLT (2019).
> [2] Serrano, Sofía and Noah A. Smith. “Is Attention Interpretable?” ACL (2019).

---

### Official Review · AnonReviewer3 · 2019-10-22
**Official Blind Review #3**

**Rating:** 6

**Review:**

This paper investigates the degree to which we might view attention weights as explanatory across NLP tasks and architectures. Notably, the authors distinguish between single and "pair" sequence tasks, the latter including NLI, and generation tasks (e.g., translation). The argument here is that attention weights do not provide explanatory power for single sequence tasks like classification, but do for NLI and generation. Another notable distinction from most (although not all; see the references below) prior work on the explainability of attention mechanisms in NLP is the inclusion of transformer/self-attentive architectures.

Overall, this is a nice contribution that unifies some parallel lines of inquiry concerning explainability, and attention mechanism for different NLP tasks. The contrast in findings between single sequence (classification) and other NLP tasks and the implications for explainability is interesting, and provides a piece missing from the analyses conducted so far.

The main issue I have with the paper is that I'm not sure I follow the rationale in Section 4 arguing that because attention "works as a gating unit", it follows that we cannot view single sequence tasks "as attention". Can the authors elaborate on why the conclusion follows from the premise? In other words, why is attention inherently *not gating*? This seems like an interesting connection, but again I do not see why attention acting as a gate implies that we should not view it as attention at all. Perhaps the authors could elaborate on their reasoning.

Some additional references the authors might consider. Note that the latter two papers, I think, would seem to broadly disagree regarding self-attention and tasks aside from single sequence (classification):
- "Fine-grained Sentiment Analysis with Faithful Attention": https://arxiv.org/abs/1908.06870
- "Interrogating the Explanatory Power of Attention in Neural Machine Translation": https://arxiv.org/abs/1910.00139
- "On Identifiability in Transformers": https://arxiv.org/abs/1908.04211

**Experience Assessment:**

I have published one or two papers in this area.

**Review Assessment: Checking Correctness Of Derivations And Theory:**

I assessed the sensibility of the derivations and theory.

**Review Assessment: Checking Correctness Of Experiments:**

I assessed the sensibility of the experiments.

**Review Assessment: Thoroughness In Paper Reading:**

I read the paper thoroughly.

---

> ### Author Response · Authors · 2019-11-07
> **Clarifications**
>
> We thank the reviewer for constructive feedback.
> In Section 4, unlike the prior works [1,2] on attention interpretability who have simply reported the observation that attention weights are not interpretable, we are providing an explanation behind that observation. By showing attention in single sequence tasks as a gating unit we want to distinguish the attention mechanism used for text classification from attention used in tasks like NLI and NMT for which it was originally proposed [3,4]. The main distinguishing factor is that in the former the attention weight for a given token is computed only based on its own representation whereas in the latter case the attention weights are based on another part of the input. We want to highlight that this simple difference is the reason behind the different nature of attention weights in both cases. In single sequence tasks, for each token attention mechanism is learning scalar to weight its representation whereas for pair sequence and generation tasks the weights are being learned with respect to some other part of the input for deciding the importance of the token. Therefore, in the latter case, the weights actually represent the reasoning behind the model’s prediction and are more critical for its performance. We will be happy to elaborate further on this in case it is not clear.
>
> Thank you for pointing out additional references. We will surely include a discussion on them in the updated version of our paper. "Interrogating the Explanatory Power of Attention in Neural Machine Translation" investigates several ways of perturbing attention weights in the NMT model of which Permute and Uniform are common with us. Please note that the authors have reported that with Permute and Uniform only 6% and 11% words are retained from the original output of the model which is quite low thus our results are consistent with what they have reported. In "On Identifiability in Transformers", authors have demonstrated that multiple attention distribution exists for the same output. However, in our work, we are analyzing that how randomly permuting attention weights in self-attention based models affect the overall prediction of the model.
>
> [1] Jain, Sarthak and Byron C. Wallace. “Attention is not Explanation.” NAACL-HLT (2019).
> [2] Serrano, Sofía and Noah A. Smith. “Is Attention Interpretable?” ACL (2019).
> [3] Rocktäschel, Tim, Edward Grefenstette, Karl Moritz Hermann, Tomás Kociský and Phil Blunsom. “Reasoning about Entailment with Neural Attention.” CoRR abs/1509.06664 (2015): n. Pag.
> [4] Bahdanau, Dzmitry, Kyunghyun Cho and Yoshua Bengio. “Neural Machine Translation by Jointly Learning to Align and Translate.” CoRR abs/1409.0473 (2014): n. pag.

---

### Official Review · AnonReviewer2 · 2019-10-25
**Official Blind Review #2**

**Rating:** 1

**Review:**

CONTRIBUTIONS:
I use (unqualified) “self-attention” to refer to attention of tokens in a sequence to other tokens in the same sequence, as described by [some corrected version of] Eq (1) and the paragraph following it (citing Bahdanau et al. 2015). This contrasts with “Transformer self-attention” and “cross-sequence attention”.
C1. Self-attention is gating. (Prop. 4.1, Sec. 4)
C2. Gating cannot provide explanations in the way that attention is alleged to do.
C3. Single input sequence models deploy only self-attention, so by C1-C2, attention cannot be used for explanation of these models; attention in two input sequence models is not limited to self-attention and is not equivalent to gating, so attention can provide explanations in these models.
C4. Outputs are sensitive to alteration of attention weights for two-sequence but not one-sequence models. (Tables 2-3 vs. 1; Figs. 2-3)
C5. Human judgments (“manual evaluation”) of the intuitive importance, for the output, of items with highest attention weights show that such intuitive importance is found for both one- and two-sequence models. (Fig. 5)
RATING: Reject
REASONS FOR RATING (SUMMARY). Because the modeling experiments closely follow previous work, the primary contribution rests on the account provided of why explanation-by-attention works only sometimes --- on the basis of the Proposition identifying attention with gating. But the reasoning and the math as presented are problematic. There is a worthwhile contribution from the human judgment experiment, but this is not sufficient to overcome the weakness with the main argument of the paper. (I also have reasons to question whether one- vs two-sequence inputs is the right distinction that needs to be accounted for.)
REVIEW
C1. If this point were made in plain English, ‘attention weights and gating units both multiply activation values’, I would say, “yes, obviously”. But the point is stated as a Proposition, with a Proof, so the math given should be correct. I don’t see a way to make it correct, and that shouldn’t be the job of the reader anyway. There are several errors in the dimensions of the matrices which make the equations incoherent. Eq. (1) contains W*h, a matrix product, where the dimensions stated are W in R^{d x d’} and h in R{T x m}; these cannot be multiplied. This unclarity about matrix dimensions propagates into Prop. 4.1. In the definition of g(X), we have WX + b, where b is presumably a vector. Addition then requires that WX also be a vector, but X is stated to be in R^{n x d}, so WX cannot be a vector. Whether WX + b is actually a vector or a matrix, g(X) = c^T tanh(WX + b) is not a matrix: it is either a scalar or a vector. But this can’t be. The definition of h uses the elementwise product, which requires that both arguments have the same dimensions, so g(X) must have the same dimensions as f(X). We’re told f is the identity function, so f(X), like X, must be a matrix. Furthermore, the statement of Prop. 4.1 says that self-attention can be interpreted as *a* gating unit. By the standard definition of ‘unit’, this should mean that self-attention is a scalar.
Throughout the paper, we are never told what kind of “RNN” is being assumed. If the RNN unit contains gates, as in an LSTM or a GRU, I can imagine that the intention is for Prop. 4.1 to say that (*) “the effect of self-attention can be reproduced without attention by adjusting the weights in the gating mechanism already present in the RNN”, so that attention doesn’t increase the capacity of the model. But what I see in the paper does not convince me that (*) is true. (Because of the kind of global normalization required by the softmax, I actually suspect it is not.)
C2. I don’t see why (formally or intuitively) gating is not a legitimate candidate for explaining the behavior of networks containing gates; I would assume just the opposite, actually. How can it *not* be part of a satisfactory explanation? And why should changing a name from “attention” to “gating” have any bearing on whether it (whatever it is called) can potentially serve for explanation of network behavior?
C3. Leaving the formalism aside, I don’t see intuitively why, whatever an analysis of self-attention might entail about explanation, the same implication shouldn’t apply to straightforward (not Rocktaeschel) attention when two sequences are present. Why can’t we just treat the concatenation of the input sequences as a single input sequence, as standardly done for example for the Transformer? If the formal content of Prop. 4.1 were clear, perhaps this could be justified, but it is simply asserted without justification in the proof that “the same reduction does not hold in the case of pair sequence”.
C4. Claims C1-C3 attempt to give an account for why various tests of the explanatory role of attention turn out positive for two-sequence but not one-sequence tasks, a pattern previously reported and verified with new results in the present paper. I fear however that one- vs two- is not the correct empirical generalization about attention that one should try to account for. Messing about with attention weights would not be expected to alter outputs if the output is determined by the input considered as a bag of words. And there is a troubling confound in the tasks compared: the one-sequence tasks are sentiment and topic classification, where a BOW model should do very well – and I suspect that is the real reason why these tasks don’t show strong sensitivity to attention weight distribution. But the two-sequence tasks are NLI and QA, where (ideally) BOW models should not do nearly so well: paying attention to the right tokens should be important. The same is true of translation. So the confound in the tasks examined leaves it undetermined whether the crucial factor to account for is one- vs two-sequence or BOW-friendly vs BOW-unfriendly tasks.
C5. Put together, the human judgments and the sensitivity-to-altering-attention-weights seem to indicate that attention tends always to be allocated to intuitively important tokens, and that matters for the output of the two-sequence models but not for the one-sequence models. This is what we’d expect if attention is always being allocated appropriately, but for BOW-friendly tasks that doesn’t make much difference.


**Experience Assessment:**

I have published one or two papers in this area.

**Review Assessment: Checking Correctness Of Derivations And Theory:**

I carefully checked the derivations and theory.

**Review Assessment: Checking Correctness Of Experiments:**

I assessed the sensibility of the experiments.

**Review Assessment: Thoroughness In Paper Reading:**

I read the paper thoroughly.

---

> ### Author Response · Authors · 2019-11-07
> **Clarifications**
>
> We thank the reviewer for constructive feedback. We firmly believe that the reviewer’s concerns can be easily handled through additional clarifications in the updated version of the paper. We hope the reviewer will agree. We provide further details below.
>
> C1: We apologize for the inconsistency in the dimensions in Equation 1. Please note that the mistake is in stating the dimensions of W which should be R^{d’ x m}. With this change, the entire Equation (1) becomes consistent. We again apologize for the confusion in Prop. 1. We have corrected the propagated inconsistency in the updated version of the paper and have explicitly mentioned all the dimensions. We clarify that the gating unit is similar to attention as it also computes scalar weights over tokens in the sentence. g(x) is a scalar weight for a token and its dimension is not the same as x (which is a vector).
>
> Throughout the paper, we use LSTM in all our models. Adding a gating unit to models for single sequence tasks does increase the number of parameters in the model and thus help to improve the performance especially for large datasets like YELP and IMDB. However, unlike the attention weights in other kinds of tasks, they do not represent reasoning behind the model’s prediction and thus are not critical for the model’s performance.
>
> C2: Unlike the prior works [1,2] on attention interpretability who have simply reported the observation that attention weights are not interpretable, we are providing an explanation behind that observation. By showing attention in single sequence tasks as a gating unit we want to distinguish the attention mechanism used for text classification from attention used in tasks like NLI and NMT for which it was originally proposed [3,4]. The main distinguishing factor is that in the former the attention weight for a given token is computed only based on its own representation whereas in the latter case the attention weights are based on another part of the input. We want to highlight that this simple difference is the reason behind the different nature of attention weights in both cases. In single sequence tasks, for each token attention mechanism is learning a scalar to decide its importance independent of other tokens whereas for pair sequence and generation tasks the weights are being learned with respect to some other part of the input for deciding the importance of the token. Therefore, in the latter case, the weights actually represent the reasoning behind the model’s prediction and are more critical for its performance.
>
> C3: Please note that in our Bahdanau et al. attention-based model for pair sequence tasks the proposition 4.1 does not hold because there the attention weight for a token in the premise also depends on the encoding of the hypothesis given by Bi-RNN based encoder. h(x) = f(x) . \sigma(g(x)) does not hold because the second term is a function of the hypothesis as well. So, in pair sequence and generation tasks the equation becomes h(x,y) = f(x) . \sigma(g(x,y)) which is not the gating unit proposed by Dauphin et al. (2017). We apologize for not explicitly specifying this in the paper.
>
> C4 and C5: BOW-friendly and BOW-unfriendly tasks can be another possible explanation behind the different nature of attention weights. However, it is not very clear how different tasks can be categorized into these two classes as even NLI can be solved using a model which utilizes BOW features from the premise and hypothesis for identifying entailment. In our work, we have given another explanation, basic idea behind which is that if the attention weights are computed with respect to another part of the input then they reflect the model’s reasoning and are critical for the model’s output otherwise they are not. In the case of single-sequence tasks, that does not hold, therefore, there attention mechanism can be reduced to gating and altering weights doesn’t have a substantial impact on the output. However, in the case of pair sequence tasks say NLI, attention weights over premise are computed based on the hypothesis and over the source sentence in NMT based on the current hidden state. Therefore, attention weights in those tasks are more meaningful and critical.
>
> [1] Jain, Sarthak and Byron C. Wallace. “Attention is not Explanation.” NAACL-HLT (2019).
> [2] Serrano, Sofía and Noah A. Smith. “Is Attention Interpretable?” ACL (2019).
> [4] Rocktäschel, Tim, Edward Grefenstette, Karl Moritz Hermann, Tomás Kociský and Phil Blunsom. “Reasoning about Entailment with Neural Attention.” CoRR abs/1509.06664 (2015): n. Pag.
> [5] Bahdanau, Dzmitry, Kyunghyun Cho and Yoshua Bengio. “Neural Machine Translation by Jointly Learning to Align and Translate.” CoRR abs/1409.0473 (2014): n. pag.

---

> > ### Comment · AnonReviewer2 · 2019-11-15
> > **AnonReviewer2 response**
> >
> > Thanks to the authors for their reply to my review, although I am not persuaded by it. With respect to the numbered points in my review:
> > C1. There remain problems with the math that caused me again to have to work too hard to figure out what the correct equations would be and whether they support the claimed conclusion or not.
> > For example, In the Proof of Prop. 4.1, h(x) is said to be the elementwise product of two factors; the elementwise product requires the two factors to have the same number of elements, but we are told that the first factor is an m-dimensional vector and the second factor is a scalar. This is exactly the sort of problem I identified with this part of the proof in the original review.
> > Then there is unnecessary work required by the reader to sort out  confusions in variable names:  ‘x’ in the Proof is ‘h’ in Eq. 1 and ‘h’ in the Proof is ‘h_\alpha’ in Eq. 1 and ‘x’ in the lead up to Eq 1, an input token,  has no correspondent at all in the proof, although the ‘x’ in the proof is referred to as the ‘input’.
> > Furthermore, Fig 1 b does not correspond to Eq 2 : h^y_T does not feed into c, as shown, but combines with each individual h^x_i to determine u_i which combines with c to determine \alpha_i. There are other errors that interfere with understanding the argument but I will stop here.
> > C2. Although other reviewers also questioned this claim of the paper, I did not see a satisfactory answer to the question.
> > C3. Although it is claimed that attention in one-sequence models can’t provide explanations, but by C5 human judgments confirm that one-sequence attention values do capture the intuitive importance of tokens. Attention in one-sequence models is doing its job and that is legitimate grounds for using it in explanations.
> > C4. Apparently the one-sequence tasks are easy enough, though, that it is doesn’t matter much that attention is doing a good job of identifying important tokens, because even when attention is distorted the models do almost as well. Nothing in the response to the review changes my view that the confound of one- vs two-sentence inputs with the tasks performed on those inputs invalidates the generalization that the paper is trying to explain – until further work de-confounds these factors.
> > Thus I see no reason to change my rating (1: Reject).

---

> > > ### Author Response · Authors · 2019-11-15
> > > **Further Clarifications**
> > >
> > > We request the reviewer to help us improve the paper by pointing out what needs to be corrected for removing the existing confusion in the paper. We provide clarification of the reviewer’s queries below:
> > >
> > > C1: Equation 1 is taken from Jain & Wallace, (2019) whereas Equation in Prop. 4.1 is from Dauphin et al. (2017). We have cited both the papers before stating the equations so that one can refer them for details. In the paper, we have described them in brief and we apologize for some notational overloaded which has lead to some confusion.
> > >
> > > In proposition 1 equation, we have tried to follow the variable names used by Dauphin et al. (2017) so that the readers familiar with their work could correlate with what we want to convey. We feel that using ‘h_\alpha’ and ‘h’ will be more confusing for the reader as we are trying to make a general statement which can hold for other tasks as well.
> > >
> > > We feel that both the equations are straightforward and easy to follow. However, below we describe them in detail for the clarification of the reviewer:
> > >
> > > Equation 1 is used to compute the attention weight for the ith token in the input sequence. It comprises of two parts:
> > > (a) u_i  = tanh(W h_i + b);
> > > (b) \alpha_i = (exp(u_i^T c)) / (\sum_{j}exp(u_j^T c))
> > >
> > > Dimensions:
> > > h_i \in R^{m}
> > > W \in R^{d’ x m}
> > > b, c \in R^{d’}
> > > \alpha_i \in R (a scalar)
> > >
> > > Part (a) is a simple linear layer followed by tanh non-linearity. It maps h_i to another vector u_i.
> > > Part (b) u_i^T c is the computed score for the ith token. Attention weights are obtained by taking softmax of scores over all the tokens.
> > >
> > > Equation in Proposition 1 is the gating equation proposed by Dauphin et al. (2017).
> > > h(x) = f(x) . \sigma(g(x))
> > >
> > > Dimensions:
> > > h(x), f(x), x \in R^{m}
> > > \sigma(g(x)) in R (a scalar)
> > >
> > > The equation gives h(x), which is a scaled version of f(x) based on the computed gating score \sigma(g(x)).
> > >
> > > Thanks for pointing out the corrections in Fig 1b. We will surely make the required changes in the updated version of the paper.
> > >
> > > C2: We request the reviewer to provide some reasons why the explanation is not satisfactory so that we can provide further clarification.
> > >
> > > C3: A series of prior works (Jain & Wallace, 2019; Serrano & Smith, 2019; Danish et al., 2019) have already shown that attention weights in single-sequence tasks do not provide an explanation for model’s prediction. We also support this observation through our results in Table 1 and Figures 2, 3, and 5. In response to the second query of Reviewer #1, we have given justification behind the apparent meaningfulness of attention weights in single-sequence tasks.
> > >
> > > C4: In the previous response (C3), the reviewer said that our self-attention explanation does not hold for pair-sequence tasks as they can be converted to single-sequence tasks by concatenating both the sequences and pass as input to the model. By that reasoning, pair-sequence tasks should also be easy enough like single-sequence tasks and distorting their attention weights should not affect their performance. However, this is not true.
> > >
> > > Our work shows that the conclusion of prior works that the model output remains unchanged on distorting attention weights does not hold for all types of tasks. Previous works have majorly focused on single-sequence tasks. In our work, we highlight that this does not hold for all types of tasks and provide an explanation behind our claim which is backed through a series of experiments. We request the reviewer to acknowledge this contribution. The reviewer has given other theories for justifying our observations like BOW-friendly/BOW-unfriendly, easy-tasks/tough-tasks, etc. However, such explanations need to be verified first through experiments. The reviewer should not assume them to be valid without any justification.

---

### Public Comment · ~Danish_Pruthi1 · 2019-10-02
**Clarifying a few points**

This paper addressed an interesting set of questions. I would be grateful if the authors could address a few points/questions:

1. Based on the text in Section 3.2 and the architecture in Figure 1, it appears that the hypothesis representation is only used to compute the attention weights over the premise sentence. The weighted sum of tokens in the premise sentence are further fed to the classifier for prediction. So, in the case when the attention is set to uniform (or random): is the hypothesis sentence being used at all?

If indeed the hypothesis is not used (when attention is set to uniform/random), does that (to a large degree) explain the severe performance drop with classification on sentence pairs in Table 2? What is the majority class/random class classifier’s performance on these tasks? What is a premise-only classifier’s performance?

2. Could you please elaborate on how thinking about attention mechanism as a gating operation invalidates its use as an interpretation (of important tokens) for single sentence tasks?  This leap from a technical claim (about attention and gating) to a casual one (about explanation, absent a formal definition) seems to hinge upon a missing definition and chain of reasoning.

Further, I did not follow how viewing attention as a gating unit suggests that final attention weights converge to a value (via optimization) where they are not necessarily higher for tokens deemed important. Conversely, when the equivalence does not hold true (for pair of sentences, generation task, etc.), what bearing does this inequivalence have on the convergence of attention weights to higher values for important tokens?

3. In Figure 3 (a), it seems that most points do not have a high difference in zeroed attention weights. Does this mean that for most examples, the attention was close to uniform? Also, if possible, could you please share what is the average change in JS divergence for the few points when the difference in attention weight is >= 0.6

---

> ### Author Response · Authors · 2019-10-05
> **Clarifications**
>
> Hi Danish,
> Please find the answer to your queries below:
>
> 1. For pair sequence tasks, following [5, 2] we have used the model which uses encoded hypothesis representation to compute attention weights over the premise tokens. So, as per the chosen model in Uniform/Random (Train+Infer) the hypothesis will not be used whereas in Uniform/Random (Infer) it will be utilized during training. [1] has shown that hypothesis-only is also a strong baseline for NLI task. However, as per our results, the same does not hold for the premise-only model.
>
> Hypothesis in NLI task in only being used for computing attention weights over premise tokens, that means the model has all the information required for making the correct prediction but just because the attention weights are not correct the performance is dropping substantially. The class distribution for SNLI and MultiNLI is as follows:
> SNLI: {'neutral': 3219, 'entailment': 3368, 'contradiction': 3237}
> MultiNLI: {'neutral': 3123, 'entailment': 3479, 'contradiction': 3213}
> Based on the above distribution, the majority class classifier’s performance will be around 33%. The results with Uniform/Random (Train+Infer) can be seen as a premise-only classifier’s performance.
>
> 2. Please note that all the prior works on attention interpretation [2, 3, 4] have reported the observation that attention weights are not interpretable. However, in this paper, we are providing an explanation behind that observation.
>
> The gating unit involves using the weights over the input by computing the similarity between a parameter vector and the input themselves. Thus, there is no interaction between different components of the input while computing the weights. This is in contrast to how attention was originally proposed and used -- to compute weights between two different parts of the input. And so, the gating unit is not adding any strengths of the attention layer to the model. Based on our empirical results, we found out that the attention weights tend to be non-interpretable when they are computed in isolation just based on the token representation itself. However, when attention weights are computed based on another part of the input (as in the case of pair sequence and generation tasks) then they are interpretable and changing them has a severe consequence on the model’s performance. The gating unit although it learns to give more importance to tokens relevant for prediction, as shown in Section 4.4, it learns to give a high score to both positive and negative tokens thus does not reveal the reasoning behind the model’s output.
>
> 3. On analyzing the datasets, we found out that the reason behind fewer points having a high difference in zeroed attention weights on IMDB dataset is not because the model is learning uniform attention over the input tokens (which is very unlikely as well). It is because the average text length in IMDB is around 282 whereas for BABI it is 140. Therefore, it is less likely that the difference will be high in the case of IMDB. However, we get similar results for SST dataset where the average length is around 18.
>
> As we can see in the given plot itself, the overall change in JS for IMDB is definitely less compared to change in JS on BABI and New Commentary dataset. We will try to include the exact numbers in the updated version of the paper.
>
> [1] Poliak, A., Naradowsky, J., Haldar, A., Rudinger, R., & Durme, B.V. (2018). Hypothesis Only Baselines in Natural Language Inference. *SEM@NAACL-HLT.
> [2] Jain, Sarthak and Byron C. Wallace. “Attention is not Explanation.” NAACL-HLT (2019).
> [3] Serrano, Sofía and Noah A. Smith. “Is Attention Interpretable?” ACL (2019).
> [4] Pruthi, Danish et al. “Learning to Deceive with Attention-Based Explanations.” ArXiv abs/1909.07913 (2019): n. Pag.
> [5] Rocktäschel, Tim, Edward Grefenstette, Karl Moritz Hermann, Tomás Kociský and Phil Blunsom. “Reasoning about Entailment with Neural Attention.” CoRR abs/1509.06664 (2015): n. pag.

---

### Decision · Program_Chairs · 2019-12-19

**Decision:**

Reject

**Comment:**

This paper investigates the degree to which we might view attention weights as explanatory across NLP tasks and architectures. Notably, the authors distinguish between single and "pair" sequence tasks, the latter including NLI, and generation tasks (e.g., translation). The argument here is that attention weights do not provide explanatory power for single sequence tasks like classification, but do for NLI and generation. Another notable distinction from most (although not all; see the references below) prior work on the explainability of attention mechanisms in NLP is the inclusion of transformer/self-attentive architectures.

Unfortunately, the paper needs work in presentation (in particular, in Section 3) before it is ready to be published.